# End-stage kidney disease and COVID-19 in an urban safety-net hospital in Boston, Massachusetts

**Mohamed Hassan Kamel[1☯], Hassan Mahmoud[1☯], Aileen Zhen[1‡], Jing Liu[1‡], Catherine G. Bielick[2], Anahita Mostaghim[2], Nina Lin[3], Vipul Chitalia[1,4,5], Titilayo Ilori[1], Sushrut S. Waikar[1], Ashish Upadhyay[1]***

1 Section of Nephrology, Department of Medicine, Boston Medical Center, Boston University School of Medicine, Boston, Massachusetts, United States of America, 2 Department of Medicine, Boston Medical Center and Boston University School of Medicine, Boston, Massachusetts, United States of America, 3 Section of Infectious Diseases, Boston Medical Center and Boston University School of Medicine, Boston, Massachusetts, United States of America, 4 Veterans Affairs Boston Healthcare System, Boston, Massachusetts, United States of America, 5 Global Co-creation Laboratories, Institute of Medical Engineering and Science, Massachusetts Institute of Technology, Cambridge, Massachusetts, United States of America

☯ These authors contributed equally to this work.
‡ AZ and JL also contributed equally to this work.
* ashishu@bu.edu

**Data Availability Statement:** All relevant data are within the paper and/or its Supporting information files.

## Abstract

### Introduction

End-stage kidney disease (ESKD) patients are at a high risk for Coronavirus Disease 2019 (COVID-19). In this study, we compared characteristics and outcomes of ESKD and non-ESKD patients admitted with COVID-19 to a large safety-net hospital.

### Methods

We evaluated 759 adults (45 with ESKD) hospitalized with COVID-19 in Spring of 2020. We examined clinical characteristics, laboratory measures and clinical outcomes. Logistic regression analyses were performed to investigate the associations between ESKD status and outcomes.

### Results

73% of ESKD and 47% of non-ESKD patients identified as Black (p = 0.002). ESKD patients were older and had higher rates of comorbidities. Admission ferritin was approximately 6-fold higher in ESKD patients. During hospitalization, the rise in white blood cell count, lactate dehydrogenase, ferritin and C-reactive protein, and the decrease in platelet count and serum albumin were all significantly greater in ESKD patients. The in-hospital mortality was higher for ESKD [18% vs. 10%; multivariable adjusted odds ratio 1.5 (95% CI, 0.48–4.70)], but this did not reach statistical significance.

**Funding:** The authors received no specific funding for this work.

**Competing interests:** The authors have declared that no competing interests exist.

## Conclusions

Among hospitalized COVID-19 patients, ESKD patients had more co-morbidities and more robust inflammatory response than non-ESKD patients. The odds ratio point estimate for death was higher in ESKD patients, but the difference did not reach statistical significance.

## Introduction

The novel Coronavirus disease 2019 (COVID-19) is a contagious disease caused by the severe acute respiratory syndrome coronavirus-2 (SARS-CoV-2) infection. Since being identified in December of 2019, COVID-19 has been shown to adversely affect multiple organ systems, both from direct viral injury and an exuberant aberrant inflammatory response [1, 2]. Patients with end-stage kidney disease (ESKD) have a high risk for SARS-CoV-2 infection, a high burden of co-morbidities linked with poor COVID-19 outcomes, and a high in-hospital morbidity and mortality [3–8]. Hemodialysis is also linked with chronic inflammation which might impact inflammatory response to infections [9]. On the other hand, the uremic state is associated with relative immunosuppression and an attenuated cytokine storm to some infections, conditions that could potentially be protective if an exuberant immune response and cytokine storm are major contributors for adverse outcomes in COVID-19 [10]. Thus, it remains unclear how COVID-19 disease activity, particularly the host inflammatory response, and clinical outcomes differ between hospitalized ESKD and non-ESKD patients.

Early accounts from China suggested that patients with ESKD may mostly have mild COVID-19 disease course [11]. In contrast, subsequent studies from Spain and the United States demonstrated high mortality rates in ESKD patients hospitalized with COVID-19 [12, 13]. A more recent study from a large New York health system reported higher in-hospital mortality in hospitalized ESKD patients compared to non-ESKD patients [14]. The current literature underrepresents minority populations and those with lower socioeconomic status. This knowledge gap is particularly important given that both COVID-19 and ESKD disproportionaty affect minority and under-served populations in the United States [15].

In this study, we aimed to compare demographics, clinical characteristics, laboratory measures, and clinical outcomes between ESKD and non-ESKD patients admitted with COVID-19 to an urban academic medical center in Boston, Massachusetts that primarily serves racial minorities and socioeconomically disadvantaged groups. In particular, we examined the association between inflammatory markers and outcomes in both ESKD and non-ESKD groups. Understanding the differences in the epidemiology of COVID-19 between ESKD and non-ESKD groups has public health implications, and can also provide hypothesis-generating insight into disease biology in this group that is particularly vulnerable to COVID-19 transmission.

## Methods

### Study design, setting and population

We conducted a retrospective cohort study of patients with confirmed COVID-19 infection admitted to Boston Medical Center (BMC). BMC is a 514-bed urban academic medical center and the largest safety-net hospital in the New England region of the United States. A confirmed case of COVID-19 was defined by a positive result on a reverse transcriptase-

polymerase chain reaction (RT-PCR) assay of a specimen collected on a nasopharyngeal swab specimen.

This study included data on 759 adults with a confirmed diagnosis of COVID-19 admitted to BMC from March 4, 2020 to April 30, 2020. Children under the age of 18 and kidney transplant recipients not receiving chronic maintenance dialysis treatments were excluded from the study. The study cohort included 45 patients with ESKD receiving chronic dialysis therapy (hemodialysis or peritoneal dialysis). March 4, 2020 was the day the first patient with ESKD and COVID-19 was admitted to BMC, and March and April included a period when the 2020 COVID pandemic was at its peak in Boston. All activities associated with our project were approved by the Boston University Medical Campus Institutional Review Board with waiver of informed consent to access non-anonymized patient data. Patient medical records from Boston Medical Center were accessed from May to July 2020.

## Data collection

Demographic and clinical data for patients were obtained manually from the hospital's electronic medical record using a research form in Research Electronic Data Capture software (REDCap, Vanderbilt University) and from the clinical data warehouse. Patients were followed until the end of their hospitalization. Health records were not ananymized prior to our access. Relevant data were entered in the database that did not include patient's name. Data was then subsequently analyzed anonymously. For patients with > 1 hospitalization during the study period, only data from the first hospitalization was used for analysis.

## ESKD status

ESKD (chronic treatment with hemodialysis or peritoneal dialysis) was confirmed by two study investigators (MH, HM or AZ, and AU) who performed independent adjudication of the ESKD diagnosis through manual chart review and comparison with inpatient dialysis records. Kidney transplant recipients were identified through manual hospital record review, and were not included in the ESKD group unless they were treated with maintenance dialysis. Kidney transplant recipients were also excluded from the non-ESKD group.

## Variables and study definitions

Data on patient demographics, co-morbid conditions, laboratory parameters, and clinical course in the hospital were collected through chart review and the electronic clinical data warehouse. For laboratory parameters, we collected admission and in-hospital values for white blood cell count (WBC), platelet count, D-dimer, ferritin, lactate dehydrogenase (LDH), C-reactive protein (CRP) and albumin levels for all patients, and serum creatinine values for non-ESKD patients. For clinical parameters, we collected data on in-hospital mortality, mechanical ventilation, ICU stay, supplemental oxygen requirement, ICU length of stay, and hospital length of stay. In addition, for patients with ESKD, we also collected data on presenting symptoms, chest x-ray findings and co-existing infections.

Presenting symptoms and co-existing conditions were ascertained from treating physician documentation and hospital record review. Fever was defined as a body temperature >100.4˚F. Bacterial pneumonia was defined by the presence of bacteria in the culture of sputum or bronchial secretions. Worsening oxygen requirement was defined as any need for supplemental oxygen above what was required prior to hospital admission. Chest x-ray findings were based on the formal radiology reports. Clinical outcomes (death, ICU admission, mechanical ventilation support, supplemental oxygen therapy, ICU length of stay and hospital length of stay) were ascertained by chart review.

## Statistical analysis

Demographic variables, laboratory findings and clinical outcomes were compared between ESKD and non-ESKD groups. We used chi-squared test to compare categorical variables, and unpaired two-samples t-test and Mann Whitney U test for continuous variables, as appropriate. Categorical variables were reported as counts with percentages. Continuous variables were reported as means with standard deviations or medians with interquartile ranges for normal or non-normal distribution, respectively.

We used unadjusted and adjusted logistic regression analyses to investigate the associations between ESKD status and clinical outcomes. For the multivariable logistic regression, we created four models for analysis. In model 1, we adjusted for age, sex, race and ethnicity. In model 2, we adjusted for covariates in Model 1, body mass index (BMI), hypertension, diabetes, congestive heart failure and chronic obstructive pulmonary disease. In Model 3, we adjusted for Model 1 variables and important admission laboratory parameters, including WBC count, ferritin, LDH, D-dimer and CRP. In Model 4, we adjusted for all the variables in Models 1, 2 and 3. We also investigated the associations between inflammatory markers (both admission values and the maximal change during the hospitalization) and clinical outcomes.

To handle missing data in the regression models, we performed multiple imputation by chained equations (MICE) with random forests and implemented predictive mean matching [16].

For additional analyses, we compared demographic variables, clinical presentation on admission, laboratory findings and clinical outcomes between ESKD patients who died during the hospitalization and those who survived the hospitalization.

All statistical tests were two-sided, and P-value of <0.05 was considered statistically significant. All statistical analyses were performed using R software version 3.6.2.

## Results

### Baseline characteristics

Between March 4, 2020 and April 30, 2020, a total of 45 ESKD patients on chronic dialysis (3 peritoneal dialysis and 42 hemodialysis) and 714 non-ESKD patients were admitted with COVID-19 at Boston Medical Center. ESKD patients were on average older (64.0 years vs 58.7 years, p = 0.01), more likely to self-identify as Black (73.3% vs 46.5%, p = 0.002), and have comorbidities including hypertension (95.6% vs 44.0%, p<0.001), diabetes (75.6% vs 28.3%, p<0.001), coronary artery disease (31.1% vs 7%, p<0.001), congestive heart failure (33.3% vs 2.2%, p<0.001), and chronic obstructive lung disease (13.3% vs 5.0%, p = 0.04) (Table 1). 9% of ESKD patients and 14% of non-ESKD patients were homeless.

### Laboratory markers of disease activity and inflammation

At the time of admission, WBC count, platelet count and serum albumin levels were similar in the ESKD and non-ESKD patients (Table 2). CRP levels were abnormally elevated (greater than the upper limit of the reference range of 5 mg/mL) in 96% of ESKD and 94% of non-ESKD patients, but similar between the two groups. D-dimer levels were abnormally high (> 243 mg/mL) in 71% of ESKD and 74% of non-ESKD patients, but similar between the two groups. Ferritin levels were abnormally elevated (> 209 ng/mL) in 100% of ESKD and 75% of non-ESKD patients, and, on average, approximately 6-fold higher in those with ESKD versus those without ESKD. LDH levels were abnormally elevated (> 309 U/L) in 52% of ESKD and 64% of non-ESKD patients, and, on average, modestly lower in the ESKD group.

**Table 1. Baseline sociodemographic and clinical characteristics of patients with COVID-19 presenting to a large safety-net hospital in Massachusetts between March 4, 2020 and April 30, 2020.**

| Characteristic | N (%), mean (SD), or median (IQR) | | P-value |
| --- | --- | --- | --- |
| | COVID- ESKD (N = 45) | COVID- Non ESKD (N = 714) | |
| Age in years, mean (SD) | 64.0 (12.6) | 58.7 (16.4) | 0.01 |
| Men, N (%) | 27 (60.0) | 414 (58.0) | 0.91 |
| Race, N (%) | | | |
| White | 4 (8.9) | 111 (15.5) | 0.002 |
| Black | 33 (73.3) | 332 (46.5) | |
| Other/ Not known | 8 (17.8) | 271 (38.0) | |
| Hispanic ethnicity, N (%) | 7 (15.6) | 253 (35.4) | 0.01 |
| Homelessness, N (%) | 4 (8.9) | 102 (14.3) | 0.43 |
| Smoking history, N (%) | | | |
| Current | 4 (8.9) | 73 (10.2) | 0.97 |
| Prior | 16 (35.6) | 157 (22.0) | 0.05 |
| Body mass index, kg/m$^2$, mean (SD) | 28.4 (7.0) | 30.9 (8.8) | 0.03 |
| Hypertension, N (%) | 43 (95.6) | 314 (44.0) | <0.001 |
| Diabetes Mellitus, N (%) | 34 (75.6) | 202 (28.3) | <0.001 |
| Asthma, N (%) | 2 (4.4) | 65 (9.1) | 0.43 |
| Coronary artery disease, N (%) | 14 (31.1) | 50 (7.0) | <0.001 |
| Congestive heart failure, N (%) | 15 (33.3) | 16 (2.2) | <0.001 |
| Chronic obstructive lung disease, N (%) | 6 (13.3) | 36 (5.0) | 0.04 |
| Prior kidney transplantation, N (%) | 2 (0.04) | - | - |
| Serum creatinine on admission (mg/dl), median (IQR) | - | 0.99 (0.80, 1.35) | - |
| Dialysis access, N (%) | | | - |
| AVF/AVG | 37 (82.2) | - | - |
| Central Venous Catheter | 5 (0.1) | - | - |
| Peritoneal Dialysis | 3 (0.7) | - | - |

Abbreviations: COVID-19- Coronavirus disease-19; ESKD- End Stage Kidney Disease; AVF- Arteriovenous Fistula; AVG- Arteriovenous Graft; N- Number; SD- Standard Deviation; IQR- Interquartile Range.

Despite having similar admission lab values with the exception of ferritin and LDH, ESKD and non-ESKD patients exhibited marked differences in the changes in laboratory parameters during hospitalization (Table 2, Fig 1). The average rise in WBC count, LDH, ferritin, and CRP were all significantly greater in those with ESKD. Platelet count and serum albumin decreased to a greater extent in those with ESKD compared to patients without ESKD. The increase in D-dimer levels was greater in ESKD patients, but the difference was not statistically significant.

## Major clinical outcomes

A total of 8 of 45 (18%) ESKD patients and 72 of 714 (10%) non-ESKD patients died during hospitalization (P = 0.11). Rates of mechanical ventilation, ICU-level care, and new supplemental oxygen use were similar between the two groups. Median length of stay was 7 days longer in ESKD versus non-ESKD patients (Table 3).

Table 4 shows results of multivariable adjusted logistic regression analyses comparing the risk of death, need for ICU stay, and need for supplemental oxygen use. In the fully adjusted model, ESKD patients had a non-significant 2-fold higher odds of the composite endpoint

**Table 2. Comparison of laboratory markers between ESKD and non-ESKD patients presenting to a large safety-net hospital in Massachusetts between March 4, 2020 and April 30, 2020.**

| Laboratory parameter | Reference range | Mean (SD) or Median (IQR) | | P-value |
|---|---|---|---|---|
| | | COVID- ESKD (N = 45) | COVID- Non ESKD (N = 714) | |
| **On admission** | | | | |
| White blood cell, K/UL | 4.0–11.0 | 7.5 (4.2) | 7.7 (4.6) | 0.68 |
| Platelet count, K/UL | 150–400 | 209 (95) | 235 (106) | 0.08 |
| Albumin, g/dL | 3.4–5.4 | 3.7 (0.5) | 3.7 (0.4) | 0.63 |
| Lactate dehydrogenase, U/L | 171–308 | 310 (237, 384) | 346 (276, 457) | 0.02 |
| Ferritin, ng/mL | 26–209 | 2593 (1351, 3265) | 450 (206, 988) | <0.001 |
| C-reactive protein, mg/L | 0–5 | 67 (25, 148) | 73 (31, 138) | 0.72 |
| D-dimer, ng/mL | < 243 | 395 (238, 915) | 371 (234, 746) | 0.80 |
| **Magnitude of change during hospitalization relative to admission values** | | | | |
| White blood cell, K/uL | - | + 1.7 (0.0, + 6.3) | + 0.6 (0.0, + 2.8) | 0.03 |
| Platelet count, K/uL | - | -33 (-11, -58) | -9 (0, -36) | 0.001 |
| Albumin, g/dL | - | -0.8 (-0.5, -1.1) | -0.5 (-0.3, -0.8) | <0.001 |
| Lactate dehydrogenase, U/L | - | + 125 (+ 8, + 244) | + 39 (0, + 132) | 0.008 |
| Ferritin, ng/mL | - | + 3363 (+ 530, + 7417) | + 85 (0, + 411) | <0.001 |
| C-reactive protein, U/L | - | + 65 (+ 17, + 187) | + 18 (0, + 66) | <0.001 |
| D-dimer, ng/mL | - | + 382 (+ 115, + 1354) | + 173 (0, + 1180) | 0.06 |

Abbreviations: COVID- Coronavirus disease-19; ESKD- End Stage Kidney Disease; n- Number; SD- Standard Deviation; IQR- Interquartile Range.

Number of missing values: White blood cells (n = 4); Platelet count (n = 4); Albumin (n = 9); Lactate dehydrogenage (n = 38); Ferritin- 32; C-reactive protein- 23; D-dimer- 10.

compared to non-ESKD patients (P = 0.17). The adjusted odds ratio for death in ESKD versus non-ESKD patients was 1.50 (95% 0.48–4.70, P = 0.49).

Table 5 shows the results of additional analyses in the entire cohort on the association of inflammatory makers and clinical outcomes (death, ICU stay and supplemental oxygen requirement). Admission CRP, admission ferritin, magnitude of rise in CRP, and magnitude

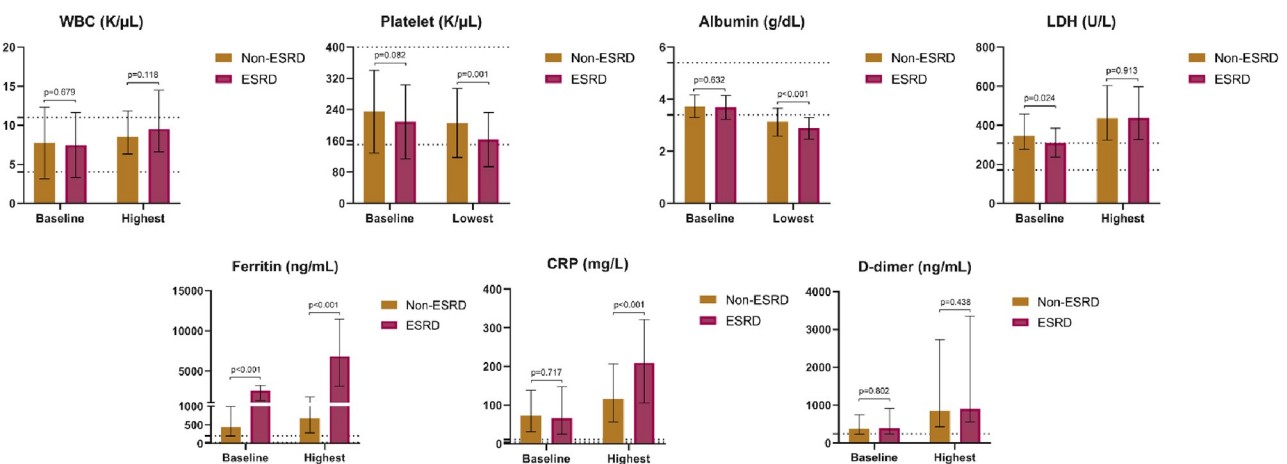

**Fig 1. Baseline and highest or lowest values for selected laboratory values in ESKD and non-ESKD groups.** Shown are means and standard deviations [for baseline White Blood Cell (WBC) count, baseline platelet count and baseline albumin], and medians and interquartile ranges [for highest WBC count, lowest platelet count, lowest albumin, lactate dehydrogenase (LDH), ferritin, C-reactive protein (CRP), and D-dimer]. Dotted line denotes the reference range.

**Table 3. Comparison of major outcomes between ESKD and non-ESKD patients.**

| Characteristic | N (%) or Median (IQR) | | P-value |
|---|---|---|---|
| | COVID- ESKD (N = 45) | COVID-Non ESKD (N = 714) | |
| Death in hospital | 8 (18) | 72 (10) | 0.11 |
| Need for mechanical ventilation | 5 (11) | 105 (15) | 0.66 |
| Need for intensive care unit | 12 (27) | 166 (23) | 0.73 |
| New supplemental oxygen requirement | 37 (82) | 516 (72) | 0.20 |
| Intensive care unit length of stay (days) * | 5 (2, 8) | 7 (2, 16) | 0.30 |
| Hospital length of stay (days) | 13 (7, 19) | 6 (3, 11) | <0.001 |

Abbreviations: COVID- Coronavirus disease-19; ESKD- End Stage Kidney Disease; N- Number; IQR- Interquartile Range.

* N = 166 for non-ESKD and 37 for ESKD.

of rise in ferritin were all significantly associated with poor clinical outcomes. The associations were strongest for the admission CRP and the magnitude of rise in ferritin.

## Characteristics associated with in-hospital mortality in ESKD

ESKD patients who died compared to those who survived were older (73.9 vs. 61.8 years, p = 0.002), more likely to have a known history of coronary artery disease (75.0% vs. 21.6%, p = 0.01), tended to present to the hospital with dyspnea (75.0% vs. 24.3%, p = 0.02), and had a higher admission ferritin (8552 ng/ml vs. 2325 ng/ml, p = 0.008) and CRP (178 U/L vs. 64 U/L, p = 0.03) levels (Table 6). Admission chest x-ray findings were similar in two groups. Death occurred a median of 6 days (interquatile rage, 2 and 15 days) after admission.

## Discussion and conclusions

In this study from the largest safety-net hospital in Massachusetts during the height of the Spring 2020 COVID-19 pandemic, we report on the clinical characteristics, laboratory measures, and clinical outcomes in hospitalized ESKD patients compared to hospitalized non-ESKD patients with COVID-19. Consistent with earlier reports, we observed that ESKD patients suffer from a higher burden of co-morbidities than non-ESKD patients with COVID-19 [12–14]. ESKD patients were also noted to have admission ferritin levels approximately

**Table 4. Multivariable adjusted logistic regression with COVID-19 with ESKD as a predictor for clinically important outcomes.**

| Outcomes | Model 1 | p-value | Model 2 | p-value | Model 3 | p-value | Model 4 | p-value |
|---|---|---|---|---|---|---|---|---|
| | OR (95% CI) | | OR (95% CI) | | OR (95% CI) | | OR (95% CI) | |
| Death | 1.58 (0.69, 3.63) | 0.28 | 2.23 (0.85, 5.87) | 0.11 | 1.19 (0.44, 3.20) | 0.73 | 1.50 (0.48, 4.70) | 0.49 |
| ICU stay | 1.16 (0.58, 2.32) | 0.68 | 1.11 (0.51, 2.43) | 0.79 | 1.24 (0.58, 2.68) | 0.58 | 1.07 (0.46, 2.49) | 0.88 |
| Supplemental Oxygen requirement | 1.74 (0.79, 3.84) | 0.17 | 2.01 (0.83, 4.86) | 0.12 | 2.03 (0.82, 5.02) | 0.13 | 2.34 (0.86, 6.39) | 0.10 |
| Death or ICU stay | 1.30 (0.68, 2.51) | 0.43 | 1.47 (0.70, 3.07) | 0.31 | 1.42 (0.68, 3.00) | 0.35 | 1.40 (0.62, 3.18) | 0.42 |
| Death or ICU stay or supplemental oxygen requirement | 1.62 (0.73, 3.58) | 0.23 | 1.91 (0.79, 4.63) | 0.15 | 1.72 (0.70, 4.26) | 0.24 | 2.02 (0.74, 5.48) | 0.17 |

Model 1: Adjusted for age, sex, race, and ethnicity.

Model 2: Adjusted for Model1, Body Mass Index, Hypertension, Diabetes, Congestive Heart Failure, and Chronic Obstructive Pulmonary Disease.

Model 3: Model1, White Blood Cell Count on admission, Lactate dehydrogenase level on admission, Ferritin level on admission, C-reactive protein level on admission, and D-dimer level on admission.

Model 4: Model1, Model2, and Model3.

Abbreviations: COVID-19- Coronavirus disease-19; ESKD- End Stage Kidney Disease; OR- Odds Ratio; CI- Confidence Interval; ICU- Intensive Care Unit.

**Table 5. Logistic regression analysis testing the association of inflammatory makers and outcomes among patients with COVID-19.**

| | All patients (N = 759) | |
| --- | --- | --- |
| | Model 1 | Model 2 |
| | OR (95% CI); p-value | OR (95% CI); p-value |
| **Outcome: Death** | | |
| Initial C-reactive protein | 8.78 (4.18, 18.47); <0.001 | 7.59 (3.59, 16.07); <0.001 |
| Initial Ferritin | 1.78 (1.28, 2.47); <0.001 | 3.60 (2.14, 6.04); <0.001 |
| Magnitude of rise in C-reactive protein | 1.37 (0.68, 2.76); 0.38 | 1.45 (0.71, 2.98); 0.31 |
| Magnitude of rise in Ferritin | 5.53 (2.87, 10.64); <0.001 | 6.26 (3.13, 12.52); <0.001 |
| **Outcome: ICU stay** | | |
| Initial C-reactive protein | 2.78 (1.86, 4.15); <0.001 | 2.71 (1.81, 4.07); <0.001 |
| Initial Ferritin | 1.78 (1.28, 2.47); <0.001 | 1.83 (1.31, 2.56); <0.001 |
| Magnitude of rise in C-reactive protein | 3.01 (1.83, 4.96); <0.001 | 3.08 (1.86, 5.11); <0.001 |
| Magnitude of rise in Ferritin | 6.98 (4.00, 12.19); <0.001 | 6.96 (3.95, 12.27); <0.001 |
| **Outcome: Supplemental Oxygen Requirement** | | |
| Initial C-reactive protein | 3.97 (2.86, 5.50); <0.001 | 3.87 (2.78, 5.40); <0.001 |
| Initial Ferritin | 2.34 (1.70, 3.23); <0.001 | 2.44 (1.76, 3.38); <0.001 |
| Magnitude of rise in C-reactive protein | 2.14 (1.20, 3.81); 0.01 | 2.15 (1.21, 3.83); 0.009 |
| Magnitude of rise in Ferritin | 4.78 (2.39, 9.56); <0.001 | 4.98 (2.47, 10.02); <0.001 |

Model 1: Adjusted for age, sex, race, and ethnicity.

Model 2: Adjusted for Model1, Body Mass Index, Hypertension, Diabetes, Congestive Heart Failure, and Chronic Obstructive Pulmonary Disease.

Abbreviations: COVID-19- Coronavirus disease-19; ESKD- End Stage Kidney Disease; OR- Odds Ratio; CI- Confidence Interval.

Number of missing values: White blood cells- 4; Platelet count– 4; Albumin-9; Lactate dehydrogenase- 36; Ferritin-32; C-reactive protein- 23; D-dimer- 107.

6-fold higher than that of non-ESKD patients. We also found higher risks of in-hospital mortality in ESKD patients with COVID-19, with odds ratios of comparable magnitude to those reported by Ng et al. in their report from 13 hospitals in the New York metropolitan area which also experienced an extremely high rate of COVID-19 hospitalizations during the Spring of 2020 (1.37 in Ng et al. vs. 1.50 in this report) [14]. Our smaller sample size limited study power and the difference in mortality between ESKD and non-ESKD groups did not reach statistical significance.

There are several features of our study that warrant emphasis. Compared to other reports in the literature, our study included a very high proportion of Black patients, a group that has been described to be at a higher risk of COVID-19 exposure, infection and severe complications [15]. Our cohort also included a substantial number of homeless individuals, which has not been studied in the context of ESKD and COVID-19.

A major finding of our study is the comparison of laboratory characteristics and their changes during hospitalization in both ESKD and non-ESKD groups. Our observation of a higher magnitude of change in laboratory markers of disease severity in the ESKD group compared to the non-ESKD group suggests a more aggressive inflammatory response in ESKD patients with COVID-19. Our study demonstrated a robust inflammatory response in ESKD patients with COVID-19, dispelling the notion that ESKD patients are potentially protected from adverse COVID-19 outcomes because of attenuated inflammatory response [11].

**Table 6. Characteristics and clinical presentation of ESKD patients who died compared to ESKD patients who survived with COVID-19 in a large safety-net hospital in Massachusetts between March 4, 2020 and April 30, 2020.**

| Characteristic | N (%), mean (SD), or Median (IQR) | | P-value |
|---|---|---|---|
| | ESKD patients who died (N = 8) | ESKD patients who survived (N = 37) | |
| **Baseline characteristics** | | | |
| Age in years, mean (SD) | 73.9 (7.5) | 61.8 (12.5) | 0.002 |
| Men, N (%) | 5 (62.5) | 22 (59.5) | 1.00 |
| Race, N (%) | | | |
| White | 0 (0.0) | 4 (10.8) | 0.17 |
| Black | 8 (100.0) | 25 (67.6) | |
| Other/ Not known | 0 (0.0%) | 8 (21.6%) | |
| Hispanic ethnicity, N (%) | 0 (0.0%) | 7 (18.9%) | 0.42 |
| Homelessness, N (%) | 0 (0.0%) | 4 (10.8%) | 0.77 |
| Smoking history, N (%) | | | |
| Current | 2 (25.0%) | 2 (5.4%) | 0.28 |
| Prior | 3 (37.5%) | 13 (35.1%) | 1.00 |
| Body mass index, kg/m$^2$, mean (SD) | 28.1 (7.0) | 28.4 (7.1) | 0.91 |
| Hypertension, N (%) | 8 (100.0%) | 35 (94.6%) | 1.00 |
| Diabetes Mellitus, N (%) | 8 (100.0%) | 2 (70.3%) | 0.19 |
| Asthma, N (%) | 0 (0.0%) | 2 (5.4%) | 1.00 |
| Coronary artery disease, N (%) | 6 (75.0%) | 8 (21.6%) | 0.01 |
| Congestive heart failure, N (%) | 0 (0.0%) | 15 (40.5%) | 0.07 |
| Chronic obstructive lung disease, N (%) | 3 (37.5%) | 3 (8.1%) | 0.10 |
| **Presenting symptoms**, N (%) | | | |
| Fever | 2 (25.6%) | 25 (67.6%) | 0.07 |
| Chills | 0 (0.0%) | 7 (18.9%) | 0.42 |
| Cough | 2 (25.0%) | 20 (54.1%) | 0.27 |
| Dyspnea | 6 (75.0%) | 9 (24.3%) | 0.02 |
| Fatigue / Myalgia | 2 (25.0%) | 11 (29.7%) | 1.00 |
| Gastrointestinal symptoms | 0 (0.0%) | 9 (24.3%) | 0.28 |
| Confusion / altered mental status | 2 (25.0%) | 5 (13.5%) | 0.41 |
| **Symptom onset to admission, days, median (IQR)** | 1 (1,1) | 1 (1,3) | 0.27 |
| **Chest x-ray finding on admission**, N (%) | | | |
| Clear | 2 (25%) | 10 (27%) | 1.00 |
| Pulmonary infiltrates | 5 (63%) | 22 (59%) | 1.00 |
| Pleural effusion | 0 | 3 (8%) | 0.96 |
| Other | 1 (13%) | 4 (11%) | 1.00 |
| **Laboratory parameters on admission** | | | |
| White blood cell in K/uL, median (IQR) | 6.3 (7.5, 10.3) | 5.8 (4.8, 8.9) | 0.09 |
| Platelet count in K/uL, mean (SD) | 210 (84) | 208 (98) | 0.96 |
| Albumin in g/dL, mean (SD) | 3.4 (0.5) | 3.8 (0.4) | 0.12 |
| Lactate dehydrogenase in U/L, median (IQR) | 346 (328, 429) | 277 (235, 357) | 0.09 |
| Ferritin, ng/mL, median (IQR) | 8552 (2952, 10132) | 2325 (1091, 2995) | 0.008 |
| C-reactive protein in U/L, median (IQR) | 178 (68, 323) | 64 (22, 105) | 0.03 |
| D-dimer in ng/mL, median (IQR) | 511 (401, 759) | 342 (231, 915) | 0.55 |
| **Magnitude of change of laboratory measures during hospitalization relative to admission values, median (IQR)** | | | |
| White blood cell, K/uL | + 1.4 (0.0, + 12.1) | + 1.7 (0.0, + 4.9) | 0.80 |
| Platelet count, K/uL | - 37 (-24, -67) | - 33 (-11, -55) | 0.59 |
| Albumin, g/dL | -0.9 (-0.5, -1.2) | -0.8 (-0.5, -1.0) | 0.72 |

*(Continued)*

**Table 6.** (Continued)

| Characteristic | N (%), mean (SD), or Median (IQR) | | P-value |
|---|---|---|---|
| | ESKD patients who died (N = 8) | ESKD patients who survived (N = 37) | |
| Lactate dehydrogenase, U/L | + 83 (+29, + 475) | + 140 (+ 12, + 242) | 0.93 |
| Ferritin, ng/mL | + 6748 (+ 2554, + 14663) | + 2510 (+ 460, + 6889) | 0.19 |
| C-reactive protein, U/L | + 78 (+ 21, + 165) | + 65 (+ 16, + 189) | 0.79 |
| D-dimer, ng/mL | + 368 (+116, + 3923) | + 382 (+ 115, + 1309) | 0.86 |
| **Other outcomes** | | | |
| Need for mechanical ventilation, N (%) | 4 (50.0%) | 1 (2.7%) | 0.001 |
| Need for ICU, N (%) | 5 (62.5%) | 7 (18.9%) | 0.04 |
| New supplemental oxygen requirement, N (%) | 8.0 (100.0%) | 29 (78.4%) | 0.35 |
| ICU length of stay (days), median (IQR) | 7.0 (2.0, 10.0) | 3.0 (2.0, 6.5) | 0.52 |
| Hospital length of stay (days), median (IQR) | 6.0 (2.0, 15.0) | 14.0 (9.0, 19.0) | 0.08 |

Abbreviations: COVID-19- Coronavirus disease-19; ESKD- End Stage Kidney Disease; N- Number; SD- Standard Deviation; IQR- Interquartile Range.

Number of missing values: White blood cells- 0; Platelet count– 0; Albumin-0; Lactate dehydrogenase- 3; Ferritin- 1; C-reactive protein- 0; D-dimer- 0.

Among markers of disease severity and inflammation, ferritin stood out as a marker that was strongly associated with poor outcomes, as well as, being disproportionately higher on admission and during hospitalization in the ESKD group than the non-ESKD group. Ferritin, a cytosolic protein best known for its role in iron storage, is also secreted by macrophages, hepatocytes and Kupffer cells during acute inflammation, especially when there is macrophage activation [17, 18]. While some degree of ferritin elevation may be expected in patients with ESKD due to their higher inflammation at baseline, the 6-fold higher admission value and the higher magnitude of in-hospital rise in ferritin observed in patients with ESKD may be related to the difference in immune response between ESKD and non-ESKD patients. It is well-established that the cytokine profile of patients with ESKD encourages a T-helper type 1 lymphocyte (Th1) over a T-helper type 2 lymphocyte (Th2) response [19]. Macrophages are one of the main effectors for Th1, and increase in Th1 over Th2 may contribute to macrophage activation and hyperferritinemia that is more pronounced in the ESKD group than in the non-ESKD group. Ferritin is composed of H and L subunits, and the H-subunit acts as immunomodulatory molecule increasing interleukin-1β, a prominent cytokine increased in patients with COVID-19 [20, 21]. Perhaps ferritin is more than a bystander but is an active participant in the hyperinflammatory response in ESKD patients.

In other studies, ESKD was also associated with higher baseline levels of interleukin-6 (IL-6) and tumor necrosis factor-alpha (TNF-alpha), two cytokines recently observed to be independently associated with higher COVID-19 severity [22, 23]. In conditions like kidney failure that is known to result in chronic immune suppression, cytokine storm syndrome due to COVID-19 is thought to occur from an over-compensatory hyper-inflammatory response mediated by IL-6 [24]. Our observation of a greater magnitude of change in laboratory markers of disease severity in the ESKD group compared to the non-ESKD group suggests a more aggressive inflammatory response in ESKD patients with COVID-19.

Our study had number of limitations, primary of which were the relatively small sample size compared and the single-center experience As the study evaluated clinical outcomes during the peak of COVID-19 pandemic in the spring of 2020, and prior to the availability of treatment options and consensus in COVID-19 treatment recommendations, there is a possibility that the availability of newer therapeutic options for COVID-19 since the completion of

our study could limit the comparison of our findings with any future observations in ESKD patients with COVID-19.

Our study also had important strengths and uniqueness to the currently available data on this new infection. Our study included a large proportion of individuals from underrepresented groups and lower socio-economic status. These groups have been disproportionately affected by COVID-19 and have not been well represented in other similar studies [12–14]. In addition, we assessed clinical outcomes and co-morbidities through manual chart review reducing the risk for adjudication errors. Unlike other studies, we also examined laboratory markers of disease severity and inflammation longitudinally throughout the hospital course, and not only on admission like many earlier studies, allowing for more robust comparisons of laboratory measures of disease activity for both ESKD and non-ESKD groups.

Future research in ESKD patients with COVID-19 is needed to better define disease mechanism, complications, and therapeutic options. Our study suggested an important role of ferritin in ESKD patients with COVID-19, and the potential utility of using ferritin levels in ESKD patients in early COVID-19 inection to stratify those at risk for poor outcomes. Doing so may help identify ESKD patients who may benefit from earlier and more aggressive management, especially as more effective treatment options and strategies for COVID-19 are being rapidly developed and studied. Our findings of robust inflammatory response in ESKD could be a marker of disease severity in this group, and suggests that patients with ESKD may benefit from immune-modulatory treatments, particularly those targeting IL-6 and macrophage activation. Future research is also needed to better understand the long-term consequences of COVID-19 in ESKD. In addition to studies elucidating disease biology, future effort should also focus on better understanding COVID-19 epidemiology in socioeconomically disadvantaged and minority groups that have been disproportionately affected by the global pandemic.

In conclusion, our study in a large urban safety-net hospital in Boston, Massachusetts shows that among hospitalized patients with COVID-19, patients with ESKD on dialysis have a higher burden of co-morbidities and a more robust inflammatory response than non-ESKD patients. While limited by study power, there was also a suggestion of worse clinical outcomes in the ESKD group. Ferritin was disproportionately higher in the ESKD group and linked with poor clinical outcomes, suggesting that disease mechanisms involving ferritin may play an important role in ESKD patients with COVID-19. Taken together, our results highlight the need for continued research in ESKD patients with COVID-19 to better define disease mechanism, complications and therapeutic options in this uniquely vulnerable patient group.

## Supporting information

**S1 File. Master dataset used for analysis.**
(XLSX)

## Author Contributions

**Conceptualization:** Mohamed Hassan Kamel, Hassan Mahmoud, Aileen Zhen, Vipul Chitalia, Titilayo Ilori, Sushrut S. Waikar, Ashish Upadhyay.

**Data curation:** Mohamed Hassan Kamel, Hassan Mahmoud, Aileen Zhen, Jing Liu, Catherine G. Bielick, Anahita Mostaghim, Nina Lin, Titilayo Ilori, Sushrut S. Waikar, Ashish Upadhyay.

**Formal analysis:** Mohamed Hassan Kamel, Hassan Mahmoud, Jing Liu, Titilayo Ilori, Sushrut S. Waikar, Ashish Upadhyay.

**Investigation:** Mohamed Hassan Kamel, Hassan Mahmoud, Catherine G. Bielick, Anahita Mostaghim, Nina Lin, Vipul Chitalia, Titilayo Ilori, Sushrut S. Waikar, Ashish Upadhyay.

**Methodology:** Mohamed Hassan Kamel, Hassan Mahmoud, Aileen Zhen, Jing Liu, Catherine G. Bielick, Anahita Mostaghim, Nina Lin, Vipul Chitalia, Titilayo Ilori, Sushrut S. Waikar, Ashish Upadhyay.

**Project administration:** Ashish Upadhyay.

**Resources:** Ashish Upadhyay.

**Supervision:** Ashish Upadhyay.

**Validation:** Mohamed Hassan Kamel, Hassan Mahmoud, Jing Liu, Sushrut S. Waikar, Ashish Upadhyay.

**Visualization:** Mohamed Hassan Kamel, Hassan Mahmoud, Aileen Zhen, Jing Liu, Vipul Chitalia, Titilayo Ilori, Sushrut S. Waikar, Ashish Upadhyay.

**Writing – original draft:** Mohamed Hassan Kamel, Hassan Mahmoud, Aileen Zhen, Jing Liu, Ashish Upadhyay.

**Writing – review & editing:** Mohamed Hassan Kamel, Hassan Mahmoud, Aileen Zhen, Jing Liu, Catherine G. Bielick, Anahita Mostaghim, Nina Lin, Vipul Chitalia, Titilayo Ilori, Sushrut S. Waikar, Ashish Upadhyay.

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
