## [Decision Letter · Decision Letter 0]

15 Apr 2021

PONE-D-21-09457

End-stage kidney disease and COVID-19 in an urban safety-net hospital in Boston, Massachusetts

PLOS ONE

Dear Dr. Upadhyay,

Thank you for submitting your manuscript to PLOS ONE. After careful consideration, we feel that it has merit but does not fully meet PLOS ONE’s publication criteria as it currently stands. Therefore, we invite you to submit a revised version of the manuscript that addresses the points raised during the review process.

We look forward to receiving your revised manuscript.

Kind regards,

Bhagwan Dass, MD

Academic Editor

PLOS ONE

Journal Requirements:

Thank you for providing the date(s) when patient medical information was initially recorded. Please also include the date(s) on which your research team accessed the databases/records to obtain the retrospective data used in your study.

Please list the exclusion criteria used for selecting patients in your methods section.

In your ethics statement in the Methods section and in the online submission form, please provide additional information about the data used in your retrospective study. Thank you for stating that "Data was analyzed anonymously and participant consent was not required." Please clarify whether all data were fully anonymized before you accessed them.

Reviewers' comments:

Reviewer's Responses to Questions

**Comments to the Author**

1. Is the manuscript technically sound, and do the data support the conclusions?

Reviewer #1: Yes

Reviewer #2: Yes

2. Has the statistical analysis been performed appropriately and rigorously? 

Reviewer #1: Yes

Reviewer #2: I Don't Know

3. Have the authors made all data underlying the findings in their manuscript fully available?

Reviewer #1: Yes

Reviewer #2: Yes

4. Is the manuscript presented in an intelligible fashion and written in standard English?

Reviewer #1: Yes

Reviewer #2: Yes

5. Review Comments to the Author

Reviewer #1: You are correct in pointing out the limitation of your study given the small sample size of the ESKD cohort. I would revise your conclusions and discussion. You data, Table 4, suggests that when you control for co-morbid conditions such as diabetes, hypertension and CHF, ESKD is not an independent risk factor for a poor outcome. In addition, in my experience, ESKD patients tend to have higher ferritin levels (as an inflammatory marker) in general so the higher levels on admission may be difficult to interpret though the greater rise in ferritin along with other inflammatory markers in the ESKD cohort with poorer outcomes makes sense.

Reviewer #2: Smaller cohort likely responsible for not being able to show higher mortality in ESKD patients compared to non-ESKD patients with COVID19 as seen in similar studies with larger cohort, otherwise, good data collection.

6. PLOS authors have the option to publish the peer review history of their article (what does this mean?). If published, this will include your full peer review and any attached files.

Reviewer #1: **Yes: **Anthony M. Valeri, MD

Reviewer #2: No

---

## [Author Response · Author response to Decision Letter 0]

7 May 2021

Response to the reviewers

We want to thank the academic editor and reviewers for their careful consideration of our manuscript titled “End-stage kidney disease and COVID-19 in an urban safety-net hospital in Boston, Massachusetts” PONE-D-21-09457. We appreciate the constructive feedback for clarification and improvement. Please find our responses to the comments below:

Responses to comments by the Academic Editor:

Response: We have reviewed the style requirements and have made necessary changes to the manuscript. 

2. Thank you for providing the date(s) when patient medical information was initially recorded. Please also include the date(s) on which your research team accessed the databases/records to obtain the retrospective data used in your study.

Response: Thank you for your comment. All activities associated with our project were approved by the Boston University Medical Campus Institutional Review Board with waiver of informed consent to access non-anonymized patient data. Patient medical records from Boston Medical Center were accessed from May to July 2020. The sentence at the end of the second paragraph of the methods section has been amended to include the above dates. 

3. Please list the exclusion criteria used for selecting patients in your methods section.

Response: Thank you for your comment. Our study included data on 759 adults with a confirmed diagnosis of COVID-19 admitted to Boston Medical Center from Mary 4, 2020 to April 30, 2020. We excluded children under the age of 18 and kidney transplant recipients not receiving chronic maintenance dialysis treatments. We have added a sentence in the second paragraph of the methods section to clarify our exclusion criteria. 

4. In your ethics statement in the Methods section and in the online submission form, please provide additional information about the data used in your retrospective study. Thank you for stating that "Data was analyzed anonymously and participant consent was not required." Please clarify whether all data were fully anonymized before you accessed them.

Response: After obtaining the waiver of informed consent from the Institutional Review Board, demographic and clinical information were extracted directly from the electronic health records us. Health records were not anonymized prior to our access. Relevant data were entered in the database that did not include patient’s name. Data was then subsequently anonymously analyzed. We added three sentences in the section on “Data collection” to clarify the above points. 

Response: We have reviewed our reference list. No retracted paper has been included as a reference and we have not changed the reference list. 

Responses to comments by Reviewer 1- Anthony M. Valeri MD:

Reviewer #1: You are correct in pointing out the limitation of your study given the small sample size of the ESKD cohort. I would revise your conclusions and discussion. You data, Table 4, suggests that when you control for co-morbid conditions such as diabetes, hypertension and CHF, ESKD is not an independent risk factor for a poor outcome. In addition, in my experience, ESKD patients tend to have higher ferritin levels (as an inflammatory marker) in general so the higher levels on admission may be difficult to interpret though the greater rise in ferritin along with other inflammatory markers in the ESKD cohort with poorer outcomes makes sense.

Response: We want to thank Dr. Valeri for his comments. To address the reviewer’s comments, we have made following changes to the manuscript:

a. In the conclusion section of the abstract, we have changed the sentence “The odds ratio of death was higher in ESKD patients, and consistent with the reports from other cohorts.” to “The odds ratio point estimate for death was higher in ESKD patients, but the difference did not reach statistical significance.”

b. In the first paragraph of the discussion and conclusions section, we have clarified that our observation of a higher mortality in ESKD patients, while comparable in magnitude to the earlier report from New York, did not reach statistical significance.

c. We agree with the reviewer that it is difficult to solely interpret higher admission ferritin level in patients with ESKD. Therefore, we have changed the sentence in the 4th paragraph of the discussion and conclusions section from, “The high ferritin seen in ESKD patients may be related to the difference in immune response between ESKD and non-ESKD patients.” to “While some degree of ferritin elevation may be expected in patients with ESKD due to their higher inflammation at baseline, the 6-fold higher admission value and the higher magnitude of in-hospital rise in ferritin observed in patients with ESKD may be related to the difference in immune response between ESKD and non-ESKD patients.”

Responses to comments by Reviewer 2:

Reviewer #2: Smaller cohort likely responsible for not being able to show higher mortality in ESKD patients compared to non-ESKD patients with COVID19 as seen in similar studies with larger cohort, otherwise, good data collection.

Response: We want to thank the reviewer for the comment. We have acknowledged the limitation of our small sample size for the mortality outcome. However, unlike other published cohorts that have only examined admission laboratory parameters, we have examined the changes in laboratory makers of disease severity and inflammation during hospital stay, allowing for more robust comparisons of disease activity for both ESKD and non-ESKD groups. In addition, the examination of the trajectory of laboratory makers also enabled us to provide hypothesis-generating insight into COVID-19 disease biology in ESKD patients.

---

## [Editor Report · Decision Letter 1]

20 May 2021

End-stage kidney disease and COVID-19 in an urban safety-net hospital

in Boston, Massachusetts

PONE-D-21-09457R1

Dear Dr. Upadhyay,

We’re pleased to inform you that your manuscript has been judged scientifically suitable for publication and will be formally accepted for publication once it meets all outstanding technical requirements.

Kind regards,

Bhagwan Dass, MD

Academic Editor

PLOS ONE
---

## [Editor Report · Acceptance letter]

26 May 2021

PONE-D-21-09457R1 

End-stage kidney disease and COVID-19 in an urban  safety- net hospital in Boston, Massachusetts 

Dear Dr. Upadhyay:

I'm pleased to inform you that your manuscript has been deemed suitable for publication in PLOS ONE. Congratulations! Your manuscript is now with our production department. 

Kind regards, 

on behalf of

Dr. Bhagwan Dass 

Academic Editor

PLOS ONE